# Conspiracy beliefs and negative attitudes towards outgroups in times of crises: Experimental evidence from Germany

Rebecca Endtricht[1], Eylem Kanol [2]*

1 Department of Criminology, University of Hamburg, Hamburg, Germany, 2 WZB Berlin Social Science Center, Berlin, Germany

* eylem.kanol@wzb.eu

## Abstract

While research on the determinants of conspiracy beliefs has been growing, there is still limited attention given to the broader consequences of conspiracy theories. This study examines the effects of conspiratorial framing on outgroup evaluations in the context of societal crises. Using an experimental design and a large representative sample of the German population, we exposed participants to conspiratorial framings of health, economic, and security crisis scenarios. The findings reveal that exposure to conspiratorial framing of crises leads to significantly more negative attitudes towards outgroups compared to control conditions. The impact is most pronounced in the security crisis treatment condition, particularly in war scenarios. Additionally, our study demonstrates the important role of political ideology, as individuals with left- as well as right-wing ideologies displayed more negative attitudes towards outgroups when exposed to conspiratorial framings of crises. These findings contribute to the literature by providing experimental evidence of the detrimental effects of conspiracy narratives on intergroup attitudes during crises.

**Data Availability Statement:** The data underlying the results presented in the study are available from https://www.fdr.uni-hamburg.de/record/14358.

## Introduction

Conspiracy beliefs are not a recent phenomenon specific to any particular culture or the digital age [1]. In Western countries, a significant rate of individuals are familiar with numerous well-known conspiracy theories, with as many as one third of the population in Germany agreeing with at least one such theory [2]. The prevalence of conspiracy beliefs tends to escalate notably during periods of crisis. This increase is often linked to heightened feelings of anxiety, uncertainty, and a perceived loss of control [1]. Wars, economic downturns, and pandemics exemplify the types of crises that have historically shaped societies and continue to exert influence today. More recently, societies have been experiencing large-scale crises as evidenced by the global COVID-19 pandemic and the ongoing war in Ukraine. These crises and their subsequent repercussions are widely recognized for engendering substantial uncertainty within societies. In response, inquiries into accountability often emerge, which prompt critical questions regarding the underlying causes of these crises and the agents involved.

**Funding:** Bundesministerium für Bildung und Forschung [grant number MOTRA- 13N15223].

**Competing interests:** The authors have declared that no competing interests exist.

Conspiracy theories can help in understanding complex situations by providing straightforward and clear explanations and directives for action. They achieve this by incorporating two crucial elements: "patterns and agency" [3]. In contexts marked by insecurity, conspiracy theories frequently posit the existence of 'evil outgroups,' who are portrayed as clandestine catalysts orchestrating crises from behind the scenes. It is worth noting that this phenomenon of scapegoating in times of societal crises is not a new phenomenon: As Emile Durkheim (as cited in Goldberg [4]) once stated, "[w]hen society suffers, it needs someone to blame, someone upon whom to avenge itself for its disappointments." This may create 'us vs them' dichotomies and erode trust between social groups that perceive themselves as victims and those that are targeted as seemingly responsible for the situation [5]. The targets of these hostilities in conspiracy theories vary widely, encompassing government institutions like secret service agencies, minority groups such as Muslims or Jews, and nation-states including the United States [1, 6, 7].

While the role of conspiracy theories in fostering hostility toward outgroups in the context of crises represents a timely and significant question, empirical research has thus far paid limited attention to this relationship. Existing observational and experimental studies have often been conducted with small, non-representative samples and have focused on a limited number of outgroups [8, 9]. To address these limitations, our study utilized an experimental design with a large representative sample of the general German population consisting of 2,171 respondents [10]. Specifically, we investigate whether and to what extent exposure to conspiratorial framing of crisis scenarios impacts individuals' negative assessments of a diverse range of outgroups. Moreover, we explore whether the impact of being confronted with conspiratorial framing of crisis scenarios on outgroup attitudes varies across the ideological predispositions of the respondents. Our study thus explores a) the variations in hostility directed towards different outgroups depending on the type of crisis encountered, b) how conspiracy beliefs of individuals contribute further to the emergence of hostility in crisis situations, and c) the role of political ideology within the context of this relationship. By employing an experimental design using a large survey sample we aim to shed light on the broader consequences of conspiracy beliefs on inter-group attitudes with a greater degree of confidence and generalizability.

## Conspiracy theories in times of crises

Social crises can be defined as periods in which the normal functioning of society and/or politics is disrupted, inducing global impacts and heightened feelings of anxiety and uncertainty. The nature of these events can be diverse–they "can be incidental (e.g., a terrorist strike) or more continuous (e.g., an economic crisis);" they may affect only parts of a society, a whole country, or can even spread across several countries and continents [3]. These periods challenge established power structures and norms, threatening individuals' values and way of life [11]. In these circumstances, fundamental human needs, including the desire to avoid uncertainty, seek security and control, and maintain a positive self-image, become particularly salient [3, 12, 13]. Such needs are particularly vulnerable in times of crises, which may lead to heightened "feelings of anxiety or uncertainty" as a result of a threat to "one's values, way of life, or even existence" [3].

Conspiracy theories, despite their lack of objective accuracy, provide a semblance of control by offering explanations [8]. Regarding the adherence to such theories, scholars often distinguish between conspiracy *belief* as an attitude and conspiracy *mentality* as a relatively stable personality trait [14, 15]. This assumption arises from the recognition that individuals possessing a conspiracy mentality often simultaneously subscribe to unrelated or even mutually

exclusive conspiracy theories [8, 15]. While conspiracy mentality should therefore be considered a stable trait, conspiracy belief relates to specific conspiracy narratives. Consequently, in light of our research questions and our experimental design, consisting of different conspiracy-oriented scenarios, we focus on 'conspiracy beliefs' instead of conspiracy mentality.

Conspiracy beliefs can serve as coping mechanisms, allowing individuals to claim a superior understanding of events, including behavioral rules that apply in respective contexts and situations [3]. In this regard, conspiracy theories can serve to fulfill desires stemming from the need for security, certainty, and accuracy of information, which are essential for comprehending the complexities of the world [8, 16]. Although information arising from conspiracy theories may not be objectively accurate, they nevertheless help people to "reclaim some sense of control [. . .] because they offer the opportunity to reject official narratives and allow people to feel that they possess a better account" [8].

Van Prooijen and Douglas note in this regard that "it is often easy to connect societal crises to the purposeful misdeeds of hostile groups, making it likely that many citizens consider the possibility of secret conspiracy formation" [11]. This is supported by Glick's observation that in times of societal frustration and discontent, majority group members become more devoted to ideologies that blame certain groups [17]. These underlying assumptions, particularly with regards to powerlessness and lack of control, have been tested in sociological and psychological research, and have been largely supported in recent years [14, 18–20].

This process "of blaming and often punishing a person or a group for a negative outcome that is due, at least in large part, to other causes" is also known as *scapegoating* [7]. This mechanism is not limited to specific groups or crises but can be directed towards a range of groups and emerge across different types of threatening situations. Regarding the types of relevant threatening situations, models of scapegoating state that "[m]undane frustrations are insufficient to spawn scapegoating movements" [17], and that situations must be extremely difficult in order to lead to scapegoating as a coping mechanism [21, 22]. In this regard, wars, economic crises, and health crises stand out as prominent triggers for widespread conspiracy narratives due to their direct and profound impacts on societies globally. *Economic crises* exacerbate resource competition and hardship, fueling heightened conspiracy belief [23] and blame towards vulnerable groups, such as Jews and immigrants [24, 25]. *Wars* instigate nationalist sentiments and create divisive 'us vs. them' mentalities, leading to scapegoating of perceived enemies or members of minority groups [26]. *Health crises* cause fear and uncertainty, driving the search for scapegoats and resulting in the stigmatization of specific groups [27, 28] (for a historical account of "othering" during pandemics see also Dionne [29]). These three types of crises—wars, economic downturns, and pandemics—represent extreme conditions conducive to the emergence of conspiracy theories. Such theories, in turn, can lead to the development of negative attitudes towards outgroups.

## Negative outgroup attitudes as a consequence of conspiracy theories

The consequences of conspiracy theories are manifold and entail both positive and negative impacts on individuals and society at large. Despite the predominantly negative perception of conspiracy theories, there are arguments supporting their potential benefits. Research indicates that conspiracy theories may foster greater transparency among political actors and state institutions, thereby holding authorities accountable in democratic processes [30]. On the individual level, conspiracy theories can serve as cognitive frameworks that assist individuals in navigating complex social and political realities [8]. However, the negative consequences of conspiracy theories are more frequently highlighted in the literature and arguably outweigh their positive effects. In particular, conspiracy theories have been linked to various detrimental

social, health-related, and political outcomes (for comprehensive overviews of these consequences, see, e.g., [8, 31]. As a consequence of the COVID-19 pandemic, research on health-related conspiracy theories and their consequences has risen during the last years. Debates surrounding the realities of the virus and the efficacy of vaccines were often entangled with conspiracy theories, highlighting a broader trend that includes the denial of scientific facts [8, 32, 33].

Research has also demonstrated that exposure to specific conspiracy theories tends to alter individuals' attitudes in alignment with the content of those theories, oftentimes increasing polarization in social and political view [8, 33]. This polarization can result in significant shifts in voting intentions and political behaviors, influencing the likelihood of non-voting and protest actions [14, 32]. Moreover, conspiracy theories may facilitate radicalization by spreading ideologies of extremist groups more easily, which is further associated with an increased acceptance of political violence [8, 32, 34]. Importantly, alongside these political ramifications, conspiracy theories can also contribute to the development of negative attitudes toward outgroups, which can manifest in prejudice, discrimination, and intergroup conflict.

The impact of conspiracy theories on prejudice towards outgroups is often explained through Integrated Threat Theory (ITT) [35]. ITT posits that individuals perceive threats to their ingroup identity during crisis situations, prompting negative evaluations of outgroups. This theory distinguishes between realistic threats, which relate to "physical or material well-being" or even "the very existence of the in-group" [36], and symbolic threats, which relate to ingroup values [35]. This approach fits the mechanism of conspiracy theories in crisis situations, which frame "the outgroup as collective conspirator that threatens the majority group's welfare or values" [6, 37]. Consequently, conspiracy theories during crises involve blaming specific outgroups, thereby contributing to intergroup conflict and reinforcing 'us-vs.-them' dichotomies. This effect is also known as *Group Polarization Effect*, which suggests that scapegoating can lead to varying degrees of hostility towards outgroups, further exacerbating already existing intergroup tensions [7, 19, 38].

Previous research has shown that outgroup devaluations can refer to various target groups, depending on the situation or narrative involved. The definition of conspiracy theories, as proposed by van Prooijen and Douglas [1], involves the attribution of responsibility to 'powerful groups,' which fits the observation that governments, elites or other politically influential entities are frequently deemed responsible for the misfortunes and problems faced by societies (see also [16, 39, 40]). However, conspiracy theories can extend beyond the government or political elite to include citizens of other countries, such as Russia, USA, or China, particularly when those countries are somehow involved in the crisis at hand [2, 41].

Additionally, minority groups are often victims of this mechanism, with Jews being a primary target. This stems from beliefs that are "built on the tradition of antisemitic conspiracy theories, where Jews [allegedly] control vast wealth and hidden networks in a search for world domination" [11, 42, 43]. In line with this, sociological survey studies have linked anti-Semitic attitudes to conspirative beliefs [44]. Another minority group that is regularly used as scapegoats are refugees. They are frequently associated with socio-economic issues within the host country, including job loss, and are perceived as abusers of welfare systems [45]. Furthermore, they are often depicted in public debates as responsible for spreading dangerous diseases [46, 47]. Relatedly, conspiracy theories such as the 'Great Replacement' and 'Eurabia' have recently become more prominent, targeting particularly refugees and members of the Muslim minority. These theories allege a secret plot by liberal elites to replace Europe's predominantly white population with refugees and Muslim immigrants, thereby posing a perceived threat to European culture.

The broad range of targeted outgroups during crises is additionally reflected by the significant amount of research that was conducted in the context of the COVID-19 pandemic as one of the most severe global crises in the last years. Studies found a rise in negative attitudes towards immigrants among a significant portion of the European population, particularly when concomitant health risks were perceived as a threat [48]. It has also been shown that the pandemic gave rise to prejudice and negative attitudes towards citizens of the EU, US, and Asian countries [38]. In contrast, some studies demonstrate that the pandemic had little or no effect on the level of hostility toward those outgroups [49, 50]. These differences are explained by Kudrnáč et al. with the mediating impact of factors such as social distrust, governmental skepticism, and political disengagement–factors arguably closely associated with conspiratorial thinking–on increasing anti-immigrant prejudice [51].

Jolley and colleagues assert that "the consequences of exposure to conspiracy theories for intergroup relations may be much broader than originally conceived, and capable of reducing more widespread intergroup tolerance" [6]. This notion hints at the *generalization effect* [21, 52], which manifests as an overall increase in outgroup prejudice, thus spreading towards other outgroups than the original scapegoats [12]. This effect has received some support by available studies showing that outgroup attitudes are linked with each other [14, 16]. However, the directionality of this relationship remains unclear for some parts. Furthermore, their specific relationship with societal crises remains understudied to date.

In summary, the literature reviewed suggests that various crises, whether economic, health-related, or security-related, are associated with feelings of anxiety and uncertainty. These conditions create a fertile environment for the emergence of conspiracy theories, which can lead to processes of scapegoating. Consequently, we argue that individuals exposed to conspiratorial narratives in the context of crises are likely to harbor more negative attitudes toward outgroups. Based on this discussion we draw the following hypotheses:

H1a: *Exposing individuals to a conspiratorial framing of an economic crisis will lead to a deterioration in their evaluations of outgroups.*

H1b: *Exposing individuals to a conspiratorial framing of a health crisis will lead to a deterioration in their evaluations of outgroups.*

H1c: *Exposing individuals to a conspiratorial framing of a security crisis will lead to a deterioration in their evaluations of outgroups.*

## Conspiracy theories and intergroup attitudes in Germany

Germany presents a compelling and relevant case for studying conspiracy theories. Its complex history, especially in the 20th century–marked by events like World War II, the Nazi regime, and the Cold War division–has deeply influenced the nation's collective memory and public discourse [53]. This historical backdrop has fostered various conspiracy theories, ranging from those related to World War II and the Holocaust to the Stasi's actions and influence in East Germany [54, 55]. In recent times, Germany's leadership in the European Union, its handling of the so-called 'refugee crisis' in 2015 and the COVID-19 pandemic, as well as its role in various international issues, particularly the conflict with Russia, have frequently been intertwined with conspiracy narratives [56, 57].

Public opinion studies highlight a significant presence of such theories across Germany, as detailed by Molz and Stiller [56]. A prominent survey study on right-wing attitudes in Germany reveals that a significant portion, approximately 46%, of the surveyed individuals hold the belief that clandestine organizations exert considerable influence over political decisions

[58]. This trend also encompasses perceptions regarding the intertwinement of media and politics, with a notable share of respondents expressing greater trust in personal intuition over expert opinion. Findings from a similar, more recent study indicate that approximately one-third of German voters subscribe to the belief that secret powers manipulate global events, with this belief showing an increase in prevalence compared to previous years [59].

Bergmann and Butter discuss how populist parties around the world often employ conspiratorial rhetoric, which resonates well with many of their supporters [60]. Taggart contends that conspiracy theories serve as a mobilization tool for populist leaders, offering simplified explanations for the challenges faced by societies [61]. Similarly, Brubaker views these as over-simplified and distorted reactions to complex global challenges like globalization, migration, pandemics, or climate change [62]. Observations by both scholars and security agencies highlight that the German populist radical-right party Alternative für Deutschland (AfD) plays an active role in the propagation of such right-wing extremist conspiracies as part of their populist narratives [63]. This situation raises concerns about the potential for further radicalization within parts of the right-wing spectrum, fueled by their own conspiracy theories and apocalyptic narratives. Relatedly, studies indicate that the AfD tends to garner disproportionately high support for conspiracy theories among its followers [64].

## Exploring the role of political ideology

The prominence of far-right conspiracy theories and the ascent of the AfD party in Germany prompts an investigation into whether individuals on the ideological extremes are more influenced by conspiratorial narratives. Drawing on Hardin''s exploration of political extremism and rationality [65], van Prooijen et al. contend that political extremists suffer from a "crippled epistemology," which leads them to view their political ideologies as the simplest and sole solutions to societal challenges [66]. Accordingly, this mindset predisposes them to adopt conspiracy theories as logical causal explanations for various societal events, including crises. A related characteristic is that conspiracy theories often accuse a small group of powerful, malevolent individuals of pursuing evil agendas at the expense of the greater good [67]. Similarly, extremist political extremists frequently target identifiable "evil" groups in their rhetoric. In a related vein, Bogatzki et al. highlight that, within the context of the COVID-19 pandemic, ideological beliefs, rather than threats induced by the pandemic, predominantly fuel prejudice [68].

Early classical research on authoritarianism posited that conspiracy beliefs are integral to authoritarian worldviews [69], while later studies have identified a particularly strong link with right-wing authoritarianism [70]. Consistent with this, numerous findings indicate a linear relationship between conspiracy beliefs and the political right, showing that conservatives and right-leaning individuals are more likely to endorse conspiracy theories [71, 72]. However, some critics argue that the strength and direction of the relationship between political orientations and conspiracy beliefs depend on the characteristics of the specific conspiracy theory, the degree to which it has been endorsed by political elites, and the particular socio-political context in which these theories are examined [73]. Relatedly, a prominent body of literature demonstrates a quadratic relationship between the direction of (unidimensional) political ideology and conspiracy beliefs, noting that individuals at the ideological extremes are more inclined to endorse conspiracy theories [66]. For instance, investigating a large sample from 26 countries, Imhoff et al. found that individuals at the extremes of the political spectrum are more likely to believe in conspiracy theories [67]. Similarly, a study conducted in Sweden by Krouwel et al. reveal that, in contrast to moderates, individuals with extreme ideological beliefs are significantly more prone to conspiracy thinking [74]. Notably, those on the political left exhibited a higher propensity for conspiracy beliefs than their counterparts on the right. Consequently, we

anticipate that respondents with more extreme political affiliations–at both left and the right–will respond more intensely to a conspiratorial framing of crises, displaying heightened negative attitudes toward outgroups.

## Research design and data

### Data and sampling strategy

The data used in this study was obtained from a representative survey conducted in Germany between March 18 and June 10, 2021, as part of the Monitoring System and Transfer Platform Radicalization (MOTRA) research cluster. The MOTRA research cluster focuses on various aspects of political extremism in Germany, including prevalence rates as well as possible reasons and risk factors for extremist attitudes in society [10]. The sampling and fieldwork for this study were conducted by the market research company Kantar GmbH. The sample was compiled in two stages: First, a sample of 121 municipalities was drawn from the entire population of municipalities in Germany. Subsequently, 6,000 addresses were randomly selected from population registration offices within these 121 municipalities. Potential respondents aged over 18 and residing in Germany were invited to participate in the survey using a pen-and-paper or online questionnaire and provided their informed written consent prior to completing the survey (see also [10]). In total, 2,171 respondents were sampled with a response rate of 36.6%. For our regression analyses, we removed all observations with missing values in the dependent and control variables, resulting in an overall sample of n = 1,972 respondents.

The average age of the respondents is 51 years (SD = 18), ranging from 18 to 95 years. Among the respondents, 953 (48%) stated that they were male, and 1019 (52%) stated that they were female. Regarding educational attainment, 315 respondents (16%) have a primary education degree or no degree, 554 respondents (28%) have a secondary education degree, and 1103 respondents (56%) have *Abitur*, the highest German high school degree. Additionally, 507 respondents (26%) have a migration background, meaning that either they or one of their parents migrated to Germany. In terms of religious affiliation, 1111 respondents (56%) stated that they were Christian, 76 respondents (4%) stated that they were Muslim, 749 respondents (38%) did not belong to any religious affiliation, and 36 respondents (2%) belonged to another religious group. Regarding the place of residence, 427 respondents (22%) resided in one of the federal states that belonged to the former German Democratic Republic. The majority of respondents (72%, 1489 individuals) participated in the survey by completing and returning the paper questionnaire via post (paper-and-pencil interview, PAPI), while 566 respondents (28%) participated online by following the link they received in the mail (computer-assisted-web-interview, CAWI). Lastly, respondents were asked to self-place themselves on an ideological scale, ranging from 1, left-wing to 10, right-wing. For the purpose of the analysis of heterogeneous treatment effects across ideology, we created a categorical variable by recoding respondents who responded with 1 to 4 as "Left" (836, 43%), 5 to 6 as "Center" (839, 44%), and 7 to 10 as "Right" (230, 12%). This skewed distribution of the political self-placement is quite typical for Germany, as demonstrated by an analysis of the *Politbarometer* over a period of 20 years [75]. Additionally, during the field phase of our survey in the first half of 2021, a preference for center-left parties in Germany was evident, with the Greens emerging as the strongest party in electoral polls [76]. The far-right party Alternative for Germany (AfD) garnered 11% in these polls, whereas around 5% of our respondents indicated a willingness to vote for the AfD (see also S2 Table in S1 File). This apparent but slight under-representation of politically right-leaning respondents in terms of ideological self-placement as well as voting intention should be kept in mind for the interpretation of our results regarding the heterogeneous treatment effects across ideology.

## Experimental design

In order to investigate the influence of conspiratorial framing in different crisis situations on individuals' attitudes towards outgroups, we conducted a randomized factorial survey experiment, which included three treatment groups and one control group. Our treatments consisted of brief vignettes suggesting conspiratorial involvement in the occurrence of three types of crises, namely, health crises (i.e., diseases), economic crises, and security crises (i.e., wars):

> [*Diseases / economic crises / wars*] are among the major problems of our time. Some people claim that certain groups are currently exploiting this and even promoting this for their own purposes. To what extent do you personally believe that certain groups are currently purposefully and actively [*spreading diseases in the world / causing economic crises / instigating wars in the world*]?

All respondents were randomly assigned to one of the four groups, with those in the control group not presented with a vignette. Overall, there were 555 (25.6%) respondents in the "diseases" treatment group, 557 (25.7%) respondents in the "economic crises" treatment group, 526 (24.2%) respondents in the "wars" treatment group, and 533 (24.6) respondents in the control group. As part of the vignette, respondents were asked whether they actually believed that certain groups (which were not explicitly named) were responsible for the respective crisis. With this wording, ideas of conspiracy belief were primed, while not hinting at specific narratives or particularly relevant groups in order to avoid framing certain conspiracy beliefs that would bias further answers. Instead, participants had to interpret the question and possible actors for themselves. This approach addresses criticism of traditional survey questions in conspiracy theory research, as the use of the term 'conspiracy theory' has been shown to potentially bias responses [8, 19]. Therefore, recent studies have adopted similar approaches that avoid using this term (e.g., [6]). The answer categories ranged from 1, *do not believe at all*, to 6, *completely believe*. The control group was not presented with this question.

Table 1 shows the distribution of respondents across the three treatment groups and the mean scores of agreements to the conspiracy belief-question (see S1 Table in S1 File for the distribution of responses on the Likert scale). Overall, respondents were more likely to believe that certain groups would purposefully instigate wars in the world ($M = 4.1$, $SD = 1.3$) than cause economic crises ($M = 3.2$, $SD = 1.4$) or spread diseases ($M = 2.5$, $SD = 1.5$). This is particularly interesting, given the fact that the survey was fielded during a global pandemic and thus in times of widespread conspiracies surrounding the origins of the virus.

## Outcome variables

After presenting the conspiratorial framing of the social crises and the conspiracy belief question, we presented a feeling thermometer to measure respondents' attitudes toward a set of relevant outgroups, which the control group answered as well [77]. Feeling thermometers have

**Table 1. Distribution of respondents across treatment groups and their mean scores on the conspiracy belief question.**

| Treatment | Mean | SD | N |
|---|---|---|---|
| Diseases | 2.52 | 1.52 | 555 |
| Economy | 3.15 | 1.40 | 557 |
| Wars | 4.14 | 1.31 | 526 |
| Control | - | - | 533 |

been widely and effectively utilized in numerous studies as a valid instrument for assessing interethnic and interreligious attitudes (e.g., [78, 79]). These studies span a diverse range of contexts and have consistently demonstrated the utility of feeling thermometers for capturing variations in attitudes across different ethnic and religious groups. Specifically, we chose to assess attitudes towards US-Americans, Chinese, Russians, Jews, refugees, and Muslims given that these groups have been identified in previous research as particularly susceptible to being blamed for crises, as discussed in our literature review. In addition to these outgroups, "Christians" were also included in the feeling thermometer item battery. However, given the ambiguity of "Christians" as an outgroup for non-Christian native Germans, we have excluded this group from our analyses. The feeling thermometer ranged from 0 (very negative) to 10 (very positive). For the purposes of this study, the thermometer scores were reversed so that in our analyses 10 indicates very negative opinions. Moreover, to ease the interpretation of our findings, we additionally present an average index (M = 3.87, SD = 2.07) by taking the mean of the thermometer scores for all presented groups. The Cronbach's alpha for these scores was 0.89, indicating high internal consistency (see also S2 Fig in S1 File for the correlation matrix). We also conducted a factor analysis using an oblique promax rotation. This analysis retained one factor and the loadings on this factor indicated a strong relationship between each variable and the factor (see S6 Table in S1 File).

## Results

For an overview of answer distributions regarding outgroup evaluations, we first visualize the mean scores on the feeling thermometer for outgroups across the treatment and control groups, as depicted in Fig 1. The mean scores in the control condition reflect the baseline evaluations of the sample. Overall, respondents without a crisis framing rated Muslims (M = 4.2), refugees (M = 3.9), and Russians (M = 3.9) most unfavorably, while Jews received the most favorable ratings (M = 2.8). It is noteworthy that respondents in the control group reported the lowest mean scores on the thermometer scale for each outgroup. Conversely, respondents exposed to security crises (i.e., "wars") consistently reported the highest mean scores, i.e. higher negative attitudes, for each outgroup. When comparing hostility levels of respondents in the control conditions with those across treatment conditions, it is notable that the 'baseline' of hostility is different for each outgroup, which appears to be subjectively 'adjusted' by respondents when being confronted with a crisis. Most clear in this regard is the result that Muslims, refugees, and Russians are generally faced with a higher base level of hostility than US-Americans or Jews, which is even more raised when these two groups are evaluated in connection with conspiracy-oriented framings of societal crises.

In order to investigate whether exposing individuals to a conspiracy-oriented framing of a crisis might negatively impact their perceptions of outgroups, we used Ordinary Least Squares (OLS) regression models, an approach that is robust to our dataset's characteristics. We opted for OLS regression over Analysis of Variance (ANOVA) due to violations of normality in our dependent variables, as indicated by Shapiro-Wilk tests (p < 0.001). OLS is known for its robustness to non-normality, particularly with large samples, making it a more appropriate choice for our study (see e.g., [80]). Moreover, given imbalances in participant ages and migration background across groups, OLS, unlike ANOVA, allowed us to directly control for age and migration background in our models. We performed t-tests comparing key variables across our treatment and control groups to assess the balance of our randomized groups. The majority of these tests revealed no significant differences, suggesting successful randomization for these characteristics (see S1 Fig in S1 File). However, we did find significant differences between treatment group 1 and the control group in terms of age and migration background.

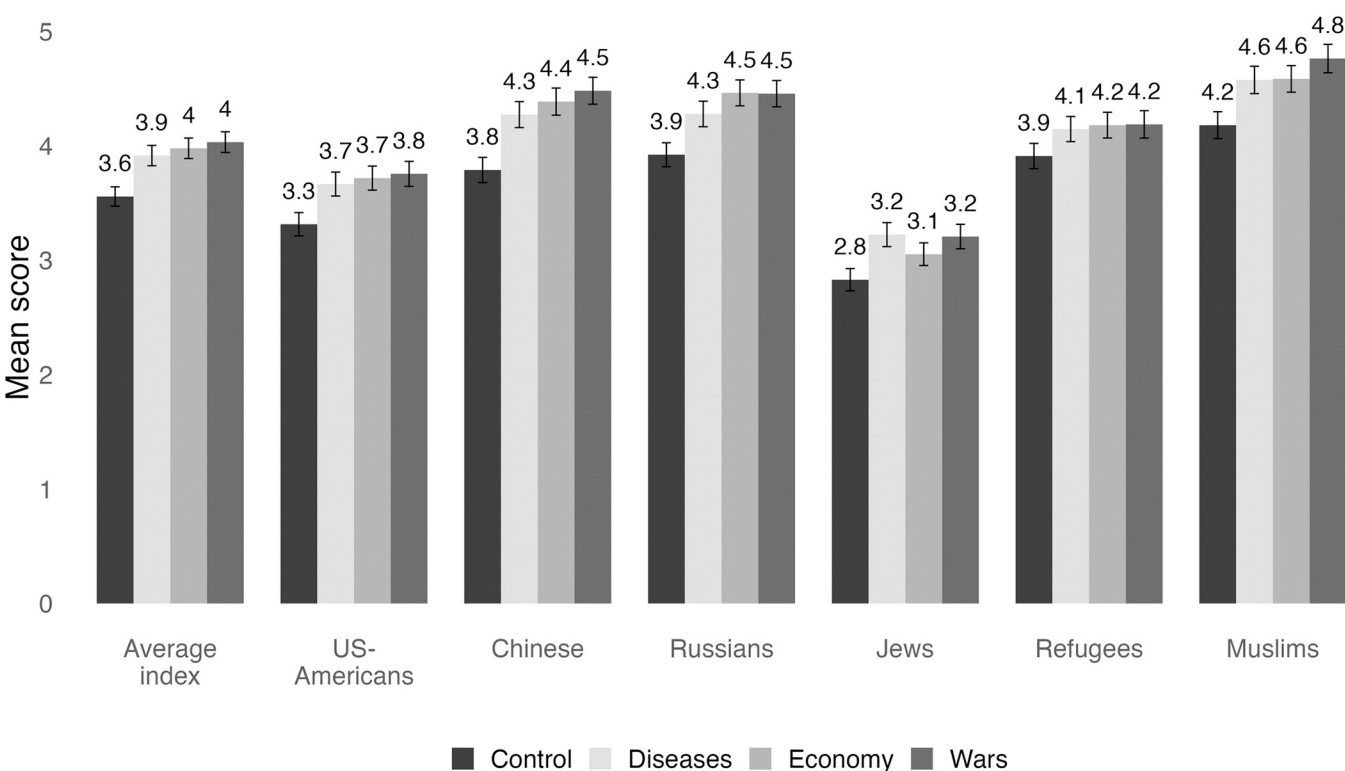

**Fig 1. Mean score on the feeling thermometer for outgroups across treatment and control groups (0, most positive ratings; 10, most negative ratings).**

Given these significant differences, we controlled for age and migration background in the OLS regression models to ensure that any observed effects are not due to these imbalances. We also estimate two alternative model specifications: one without any controls (see S3 Table in S1 File) and another with additional controls (see S4 Table in S1 File). The additional control variables include gender, level of education, religion, survey administration method, political party preference, and whether the respondent resides in a federal state that was part of the former German Democratic Republic. The overall results remain consistent across models, underscoring the robustness of our findings.

The results of the OLS regression models are presented in Table 2. Our analysis reveals a significant impact of exposure to conspiratorial framing of all crisis types on respondents' perception of outgroups, as indicated in Model 1. This significant effect is consistently observed across all individual outgroup scenarios (Models 2–7), barring attitudes towards Jews within the context of economic crises (Model 5), and towards refugees (Model 6). However, it is noteworthy that the impact of the treatment conditions on attitudes towards refugees was marginally significant at the 10% level, suggesting a trend that merits further investigation. The strongest impact is identified in the wars treatment condition. Overall, the effect of this narrative on the average rating of outgroups was 0.46 (SE = 0.13, $p < .001$). Furthermore, when looking at specific groups, the treatment effects were even more pronounced for Chinese with a coefficient of 0.69 (SE = 0.17, $p < .001$), and for Muslims with a coefficient of 0.64 (SE = 0.17, $p < .001$). Similarly high treatment effects were observed under the economic crises treatment condition regarding the evaluations of Russians, with a coefficient of 0.57 (SE = 0.16, $p < .001$), and Chinese, with a coefficient of 0.57 (SE = 0.16, $p < .001$). The effect sizes in the diseases treatment conditions were a bit lower overall (b = 0.40, SE = 0.13, $p < 0.01$), but positive and significant for all outgroups except refugees. Compared to the other two treatment groups,

**Table 2. Effect of exposing respondents to a conspiracy-oriented framing of crises on negative attitudes towards outgroups.**

| | Dependent variable: | | | | | | |
|---|---|---|---|---|---|---|---|
| | *Average index* | *US-Americans* | *Chinese* | *Russians* | *Jews* | *Refugees* | *Muslims* |
| | **(1)** | **(2)** | **(3)** | **(4)** | **(5)** | **(6)** | **(7)** |
| Treatment (ref.: CG) | | | | | | | |
| Disease | 0.40** | 0.39* | 0.50** | 0.40* | 0.45** | 0.29 | 0.48** |
| | (0.13) | (0.15) | (0.16) | (0.16) | (0.15) | (0.16) | (0.17) |
| Economy | 0.40** | 0.36* | 0.57*** | 0.57*** | 0.21 | 0.30 | 0.43* |
| | (0.13) | (0.15) | (0.16) | (0.16) | (0.15) | (0.16) | (0.17) |
| Wars | 0.46*** | 0.46** | 0.69*** | 0.52** | 0.36* | 0.28 | 0.64*** |
| | (0.13) | (0.16) | (0.17) | (0.16) | (0.15) | (0.17) | (0.17) |
| Age | 0.02*** | 0.01*** | 0.04*** | 0.03*** | 0.02*** | 0.02*** | 0.04*** |
| | (0.00) | (0.00) | (0.00) | (0.00) | (0.00) | (0.00) | (0.00) |
| Migrant background | 0.19 | 0.35** | 0.07 | -0.08 | 0.35** | 0.24 | 0.36* |
| | (0.10) | (0.13) | (0.14) | (0.13) | (0.12) | (0.14) | (0.15) |
| Observations | 1,972 | 1,972 | 1,972 | 1,972 | 1,972 | 1,972 | 1,896 |
| Adjusted R$^2$ | 0.05 | 0.01 | 0.06 | 0.05 | 0.02 | 0.02 | 0.06 |

*Note*: The Table shows OLS regression coefficients and standard errors (in parentheses). To ensure that respondents did not rate their ingroup, we removed Muslim respondents for the Muslim ratings. CG = Control Group *p<0.05

**p<0.01

***p<0.001

the effect on negative attitudes towards Jews is the highest in this group with a coefficient of 0.45 (SE = 0.15, p < 0.01).

## Amplifying effects of individual conspiracy belief

We next explore to what extent the conspiratorial framing of a crisis would particularly exacerbate negative attitudes towards outgroups among those who believe in the suggested conspiratorial framing. We estimated additional linear regression models that include the main effects for treatment conditions and the categorical conspiracy belief variable, along with their interaction, and the same set of control variables included in the previous model. To ease the interpretation of interaction effects, we plot the predicted values of the feeling thermometer (see Fig 2; for full OLS models see S5 Table in S1 File).

Our findings reveal a distinct trend: respondents who display a higher tendency towards conspiracy belief significantly express more negative attitudes towards outgroups compared to their counterparts who exhibit lower conspiracy belief levels. It is striking that the assumption of a conspiratorial involvement in a crisis leads to a higher generalized skepticism towards all presented outgroups, regardless of the crisis content. This effect is particularly strong in the 'diseases' treatment condition. In this condition, the predicted feeling thermometer score for the average outgroup measure is below 4 for respondents who did not explicitly express belief in the conspirative framing. Yet, this score leaps by over a full point, hitting over 5, among respondents who manifest high levels of conspiracy belief. The effects are particularly pronounced for the evaluation of refugees and Muslims. This observation aligns with our study's context of a global pandemic, where conspiracy theories related to the origin of the virus, the potential threats to health, and the efficacy of vaccinations were widespread and highly salient in people's minds.

The results for the security crisis treatment condition are also in the expected direction and similarly pronounced. When confronted with a conspiratorial framing of a security crisis,

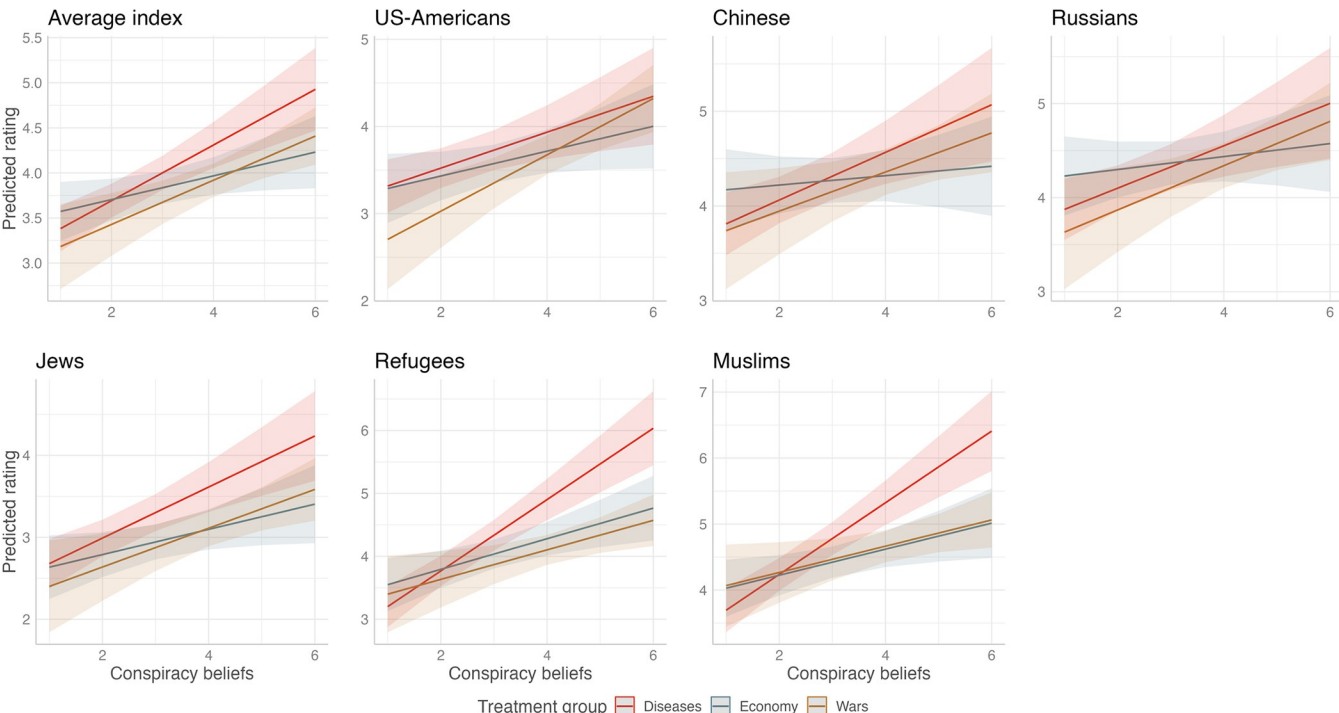

**Fig 2. Predicted ratings of outgroups on the feeling thermometer for different levels of conspiracy belief across treatment groups.** Note: The figure illustrates the predicted ratings for outgroups on the feeling thermometer, differing by levels of conspiracy beliefs across various treatment groups. Ratings derived from OLS regression models incorporating an interaction term of the treatment conditions and the conspiracy beliefs variable, while controlling for age and migration background. Note that the y-axis is unique to each subplot within the figure. The lines represent predicted values, shaded areas 95 percent confidence intervals.

individuals with higher levels of conspiracy belief exhibit significantly more negative scores on the average outgroup index. While the impact on negative attitudes towards US-Americans, Chinese, and Russians is more marked, the ratings of other outgroups do not appear to be as strongly influenced by conspiracy beliefs. Regarding the treatment condition related to economic crises, the effects are in the expected direction. However, when examining the individual outgroup ratings, the findings are not consistently significant across all groups. Respondents with higher levels of conspiracy belief exhibit significantly more negative attitudes towards Jews, refugees, and US-Americans compared to those with lower levels of conspiracy belief. On the other hand, there are no statistically significant differences in ratings for Chinese and Russians, although the direction of the differences aligns with our expectations.

Taken together, our findings largely confirm Hypotheses 1a, 1b, and 1c, offering evidence that exposing individuals to a conspiratorial framing of health, economic, or security crises does indeed negatively impact their evaluations of outgroups. However, it is also worth noting that effect sizes vary depending on the type of crisis and specific outgroups. It can be argued that outgroup hostility as a consequence of crisis perceptions is not generalized per se, but rather is adapted to groups that 'fit' to a narrative within the respective crisis scenario.

## Heterogeneous treatment effects across ideology

The last part of our analysis is concerned with heterogeneous effects of the treatment on outgroup attitudes, conditional on the ideological predispositions of respondents (Fig 3). This enables us to contribute to public debates concerning the susceptibility of proponents of right-

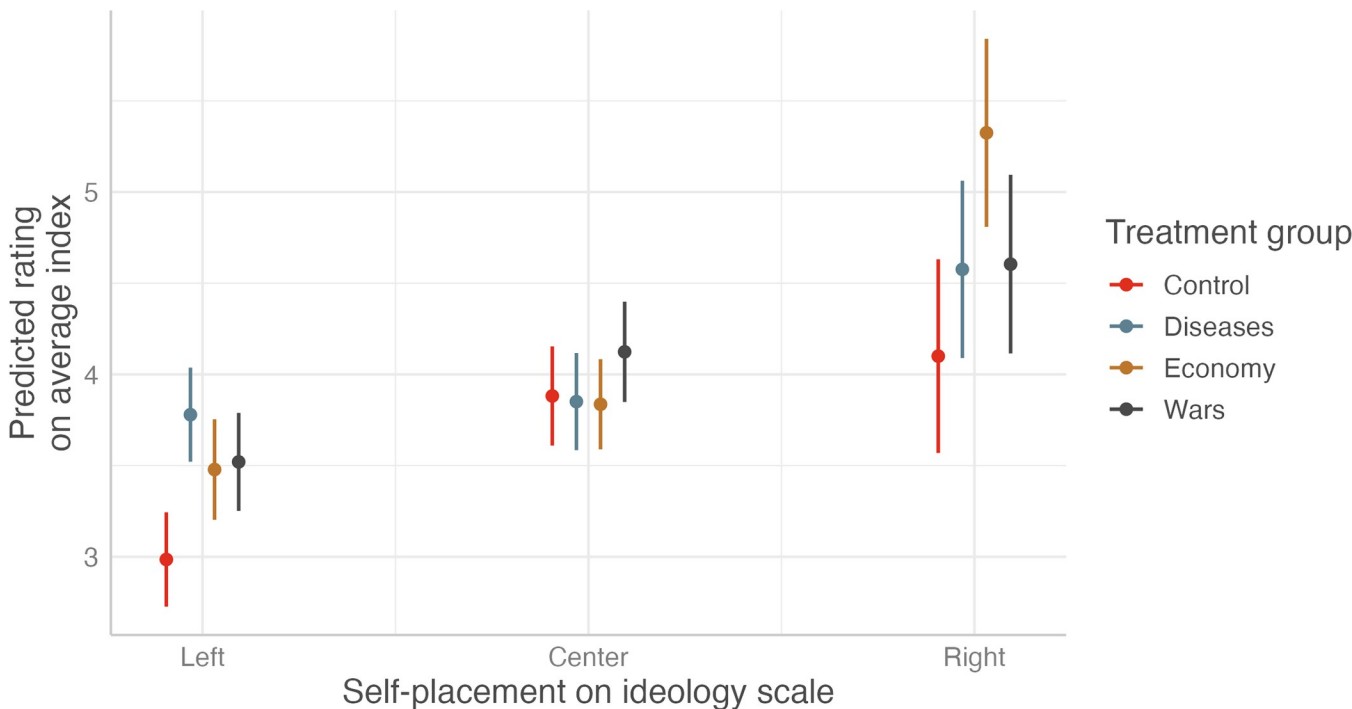

**Fig 3. Predicted ratings of outgroups on the feeling thermometer conditional on self-placement on ideology scale across treatment groups.** Note: The figure illustrates the predicted ratings for outgroups on the feeling thermometer, differing by the self-placed ideological predisposition of respondents across various treatment groups. These ratings were derived from OLS regression models incorporating an interaction term of the treatment conditions and the ideology variable, while controlling for age and migration background. Markers are predicted values, lines 95 percent confidence intervals. See S7 Table in S1 File for the full regression models.

wing ideology to conspiracy-driven narratives. Our findings indicate that individuals with right-wing predispositions are more susceptible to adopting hostile outgroup attitudes when presented with conspiracy-oriented narratives, unlike their centrist counterparts. The narrative concerning economic conditions demonstrated the most pronounced impact (1 point higher compared to the control group). Interestingly, the analysis also reveals a susceptibility among left-wing respondents to conspiratorial framing, particularly in the treatment group with the disease narratives. Overall, these observations imply that while centrist individuals demonstrate a degree of resilience to conspiracy theories, those aligned with the political extremes, both left and right, are more susceptible to them, which appears to ultimately result in more negative outgroup evaluations.

## Discussion and conclusions

This study sheds light on the powerful influence of conspiracy beliefs on outgroup perceptions in the context of health, economic, and security crises. Using a survey experiment, we demonstrate that exposing individuals to a conspiracy-oriented framing of crises can significantly deteriorate their evaluations of various outgroups. Further analysis reveals a consistent pattern in which individuals who more strongly believe in conspiratorial involvement in a crisis tend to express even more negative attitudes toward outgroups. Thus, this study unveils a notable interplay between crisis perceptions and individual conspiracy beliefs in shaping outgroup assessments.

Our study contributes to the current body of literature on conspiracy theories and intergroup attitudes in several important ways. As highlighted by van Prooijen and Douglas in their

historical review, societal crises—characterized by significant and rapid changes challenging established power structures, norms, or the status of certain groups—are often accompanied by an increased belief in conspiracy theories [11]. Essentially, these beliefs are found to be deeply linked to feelings of fear, uncertainty, or loss of control. In this regard, conspiracy theories help in satisfying the need to make sense of one's social environment amid uncertainty. In the present study, we contribute to this body of previous research by examining group-related consequences of conspiracies in the context of crises.

While previous studies have examined the relationship between conspiracy theories and prejudice, some have concentrated exclusively on a single outgroup or relied on observational data [81, 82]. Others employing experimental methods have not explored the role of conspiracies within the context of societal crises and have restricted their inquiries to specific conspiracy theories concerning selected outgroups only [6]. We expand upon this existing body of literature on conspiracy theories and prejudice by showing the broader implications of exposure to conspiracy framed crises on intergroup dynamics and social consequences of conspiracy beliefs. By employing an experimental design within a large sample of the German population, we aimed at providing a more comprehensive understanding of the direct causal effects of conspiracy theories on intergroup attitudes within the general population in the context of various societal crises.

Against the background of previous research, it could be expected that conspiracy theories "assert mainly an effect in the domain they are about" [40]. However, this suggestion is only partly supported by our results. Comparing three different crisis situations and six outgroups showed major effects across all analyses, suggesting a generalization effect of outgroup devaluation when confronted with uncertain situations. Overall, our study bridges these two lines of research on conspiracy belief and intergroup attitudes, and demonstrates that under conditions of uncertainty and crisis, conspiracy narratives can further reinforce the dichotomy between 'us' and 'them' and contribute to the development of prejudiced attitudes.

While some researchers have argued for a linear relationship between political ideology and conspiracy beliefs, with conspiracy theories being more prominent among right-wing supporters (e.g., [70, 71]), our study shows that individuals at both ends of the political spectrum are more likely to respond to conspiracy narratives with prejudiced attitudes. We observed treatment effects among both left-wing and right-wing respondents, in contrast to those who identify with the political center. This pattern aligns with the existence of a quadratic relationship, as documented in more recent studies (e.g., [65, 66, 74]). This suggests that the AfD or other far-right parties, as well as political parties on the left, can use conspiracy theories as part of their populist narratives to mobilize voters, contributing to prejudice and hostility towards outgroups.

While the observed effect sizes in our study may initially seem modest, they hold considerable practical significance. Given the scale of our feeling thermometer, ranging from 0 to 10, a shift of 0.5 in negative attitudes is indeed substantial. Importantly, this shift was prompted by only a brief exposure to our treatment—just a few sentences read within the context of a larger survey study. In real-world scenarios, individuals are often subject to conspiratorial frames in a much more pervasive and persistent manner, especially through the internet and social media [73], movies [83], or debates of political actors [84]. Therefore, the effects we observed, albeit in a controlled setting, could potentially intensify in more immersive and prolonged exposure contexts.

The importance and relevance of our findings are further underscored by the notion that "while beliefs sometimes may be flawed or even naive, they may produce behavior that has real consequences" [1]. For instance, studies have shown that conspiracy beliefs are "related to lower intentions to take part in normative political action (e.g. voting) and increased

willingness to engage with non-normative (e.g. unauthorized strike) and even violent action, both political and general" [85]. Our study thus further highlights the potential dangers of conspiracy theories. It is indeed a long step from harboring prejudiced attitudes to any form of violence, but when combined with other risk factors, strong beliefs in conspiracies might contribute to the radicalization of individuals and lead to hate crimes [86]. Consequently, it is crucial that conspiratorial narratives propagated by populist parties and other extremist groups are challenged with robust counter-narratives. In parallel, both traditional and social media platforms must enhance their efforts to monitor and curtail the dissemination of conspiracy theories.

## Limitations and further research

Our study is subject to several limitations and caveats that should be considered. Firstly, it is important to acknowledge the timing and context of the field work. The survey experiment was conducted one year after the onset of the global COVID-19 pandemic. As such, it is possible that the effects of this crisis on respondents' perceptions and attitudes differ from those of other crises, which may have been less salient during this time, thereby potentially influencing respondents' attitudes in the 'diseases' treatment condition. Another limitation relates to potential order effects in the survey questionnaire [87]. Unfortunately, the order in which the outgroups appeared was not randomized due to constraints imposed by the survey design. Therefore, we cannot rule out the possibility that the order in which the outgroups were presented may have influenced respondents' answers, potentially biasing the results.

It is worth noting that our study relied on a single outcome measure to assess negative outgroup attitudes. As a result, our experimental design did not explicitly distinguish between affective and cognitive forms of prejudice, which have been examined in previous research[6]. Future investigations could consider incorporating measures that capture both affective and cognitive dimensions to gain a more comprehensive understanding of outgroup attitudes in the context of conspiracy beliefs. To make a potential 'us-vs.-them' dichotomy more visible, it would additionally be valuable to include measures of ingroup perceptions, e.g., own perceptions of the respondents' social identity, such as religious or national identity. Additionally, understanding whether these mechanisms are part of underlying personality traits requires further investigation, including assessments of predictors of scapegoating and conspiracy beliefs, such as dichotomous thinking, feelings of anomie, or perceptions of personal, social, and political lack of control as parts of crisis coping [9].

Finally, the present study did not allow for an assessment of the enduring impact of the experimental manipulation on individuals' negative attitudes towards different groups. Previous research has indicated that conspiracy beliefs can persist for extended periods, potentially fostering sustained hostility towards outgroups. Future research employing longitudinal data can provide valuable insights into the potential long-term effects of exposure to conspiracies.

## Supporting information

**S1 File. Supporting information.**
(DOCX)

## Acknowledgments

We would like to thank Ruud Koopmans, Peter Wetzels, Ruth Dittlmann, Max Schaub, Katrin Brettfeld, Marc Helbling, Daniel Meierrieks participants of the MOTRA-Analysis Workshop 2021, participants of the MOTRA Annual Conference 2021, participants of the WZB

Migration and Diversity Colloquium, and the anonymous reviewers for their helpful comments and suggestions. We also thank Johanna Knoesel and Victoria Hoffmann for superb research assistance.

## Author Contributions

**Conceptualization:** Eylem Kanol.

**Data curation:** Rebecca Endtricht.

**Formal analysis:** Rebecca Endtricht, Eylem Kanol.

**Funding acquisition:** Eylem Kanol.

**Investigation:** Rebecca Endtricht, Eylem Kanol.

**Methodology:** Rebecca Endtricht, Eylem Kanol.

**Resources:** Rebecca Endtricht.

**Software:** Eylem Kanol.

**Supervision:** Eylem Kanol.

**Validation:** Rebecca Endtricht.

**Visualization:** Eylem Kanol.

**Writing – original draft:** Rebecca Endtricht, Eylem Kanol.

**Writing – review & editing:** Rebecca Endtricht, Eylem Kanol.

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
