## [Decision Letter · Decision Letter 0]

13 Aug 2024

PONE-D-24-24527Conspiracy beliefs and negative attitudes towards outgroups in times of crises: Experimental evidence from GermanyPLOS ONE

Dear Dr. Kanol,

Thank you for submitting your manuscript to PLOS ONE. After careful consideration, we feel that it has merit but does not fully meet PLOS ONE’s publication criteria as it currently stands. Therefore, we invite you to submit a revised version of the manuscript that addresses the points raised during the review process.

 In your revision, please pay careful attention to the following:  Both reviewers raise important concerns regarding the conceptual framework, methodological clarifications, and the integration and presentation of results. They also recommend updates to the literature, particularly emphasizing recent work in political science that tackles ideology and conspiracy beliefs. I strongly encourage you to review and incorporate recent studies published in political science to strengthen your manuscript's theoretical foundation and relevance in the current discourse.

We look forward to receiving your revised manuscript.

Kind regards,

Cengiz Erisen

Academic Editor

PLOS ONE

Journal Requirements:

2. Thank you for stating the following financial disclosure: "Bundesministerium für Bildung und Forschung [grant number MOTRA- 13N15223]"

Reviewers' comments:

Reviewer's Responses to Questions

**Comments to the Author**

1. Is the manuscript technically sound, and do the data support the conclusions?

Reviewer #1: Yes

Reviewer #2: Yes

2. Has the statistical analysis been performed appropriately and rigorously? 

Reviewer #1: Yes

Reviewer #2: Yes

3. Have the authors made all data underlying the findings in their manuscript fully available?

Reviewer #1: Yes

Reviewer #2: Yes

4. Is the manuscript presented in an intelligible fashion and written in standard English?

Reviewer #1: Yes

Reviewer #2: Yes

5. Review Comments to the Author

Reviewer #1: “Conspiracy beliefs and negative attitudes towards outgroups in times of crises: Experimental evidence from Germany” is a manuscript (MS) that rests on a survey experiment administered to a large representative sample of the general German population consisting of 2,171 respondents. It investigates whether and to what extent exposure to conspiratorial framing of crisis scenarios impacts individuals' negative assessments of a diverse range of outgroups. The findings confirm that even vaguely suggestive vignettes implying conspiratorial thinking result in significant increase in hostility to certain outgroups like Muslims or Russians. I think the MS addresses an interesting topic with appropriate methods and mostly fine presentation. It can be published after a round of revision addressing the issues below.

The experiment results are first presented visually in figure 1 as relative group means. It is then reported that there are significant imbalances between groups in terms of pretreatment cofounders; age and migration background, so there is an OLS analysis where these are controlled for. But the same OLS controls for a number of other variables, too; for whose inclusion no similar justification is offered. Since this is a randomized experiment there is no legitimate point in controlling for additional variables, whose inclusion may generate suppressor effects on behalf of the treatments, potentially leading to bias.

So I think we need to see different OLS models where

i) the treatment categories are the only predictors

ii) age and migration would be included as a predictors alongside the treatment

iii) the current OLS model with the additional controls (to give a more comprehensive understanding of the correlates of outgroup hostility, rather than to isolate the effects of the treatment).

The 1st and 3rd of these may go to some sort of appendix to save space. Please also add a measure of fit to the models. (I believe I don’t have access to the current appendix that the MS mentions, so apologies if these are already present).

The moderating effects of ideology is an important topic that can be explored with more theoretical elaboration. Treatment effects look more certain and pronounced among the left compared to the right, which is a bit at odds with the theoretical expectations (“quadratic”) laid out in the lit rev section. So this can be discussed with a few sentences.

The discussion of limitations is appropriate. However, the following sentence sounds like a non sequitur and I think should be dropped: “However, analyses of mean scores in the control group did not indicate a clear trend based on the order of the outgroups, suggesting that order effects were minimal. “Since we do not know the counterfactual of how baseline scores would fare under another random ordering we cannot infer from the observed scores whether the current ordering had any effect.

Minor: The MS has problems of precision in writing.

- At times the MS seems to display the strangely common academic mistake of conflating the concepts of “conspiracy theory” and “conspiracy”; and uses the latter to refer to the former. “Consequences of conspiracies” in the first sentence of abstract should be [consequences] “of conspiracy theories” or “of conspiracy beliefs” etc. instead. Similarly, the first subtitle that follows the intro should be sth like “conspiracy *theories* in times of crises” not conspiracies, since you are not examining conspiracies in any way.

- “a quadratic relationship between the intensity of political ideology and conspiracy beliefs concerning a range of societal issues, noting that individuals at the ideological extremes are more inclined to endorse conspiracy theories”: Sounds like this should be “Quadratic relationship between the direction of (unidimensional) political ideology…” if extremes (i.e. those with more intense ideological beliefs, left or right) are indeed more inclined.

- “heterogeneous effects of ideology” also sounds a misnomer. From the figures presented, ideology seems to have moderating effects for the treatment, resulting in heterogeneous effects of the *treatment*.

Reviewer #2: I find the manuscript titled "Conspiracy Beliefs and Negative Attitudes Towards Outgroups in Times of Crises: Experimental Evidence from Germany" to be a thought-provoking and engaging piece of research. The study looks into the consequences of conspiracy theory beliefs during crises, particularly in terms of intergroup attitudes. There is much to commend in this work, and it holds significant potential.

The article is very well-written, but some sections require further clarification. I recommend the following revisions:

1. Conceptual Clarification: The study focuses on the concept of "conspiracy beliefs" rather than "conspiracy mentality," which is a common focus in conspiracy theory research. However, the authors should clarify why they chose to use "conspiracy beliefs" instead of "conspiracy mentality." Is it because "conspiracy beliefs" are more suitable for experimental studies, or does the theoretical framework of the study necessitate this choice? A clear explanation of this decision would enhance the conceptual clarity of the paper.

2. Literature on the Consequences of Conspiracy Beliefs: While the sections “Conspiracy Theories in Times of Crises” and “Negative Outgroup Attitudes as a Consequence of Conspiracies” briefly address the literature, the paper would benefit from a more comprehensive discussion of the consequences of conspiracy beliefs. The field is rich with studies on this topic, and incorporating more of this literature could strengthen the paper. A dedicated section on the consequences of conspiracy beliefs, placed between the two aforementioned sections, could help readers better understand the study's significance.

3. Ideological Extremity and Conspiracy Beliefs: The study’s findings supporting the ideological extremity hypothesis are insightful in understanding the role of ideologies in conspiracy beliefs. However, the paper should also discuss alternative perspectives on the role of ideology in conspiracy beliefs. Citing literature that suggests right-wing supporters are more prone to conspiracy theories, or that there are no significant ideological differences in endorsing conspiracy theories, could provide a more balanced view and inform readers about different arguments on this topic.

4. Research Design and Data Section: The authors note that 12% of respondents self-identified as right-wing, compared to 43% identifying as left-wing and 44% as centrist. This distribution is intriguing and warrants further discussion. What might explain these differences? Could it be the survey sampling strategy, or perhaps it reflects the current political climate in Germany? Addressing these questions could provide valuable context for interpreting the findings.

5. Control Variables: The study controls for gender, level of education, religion, migration background, form of survey administration, and whether respondents resided in a former East German state. However, the authors might consider including party identity as a control variable, especially given the rise of the Alternative for Germany (AfD) and its influence on German politics. Including party identity (or vote choice) could offer additional insights, particularly considering the party's stance on migration.

Overall, I found this manuscript to be an engaging read, and I believe it could make a significant contribution to the field after some revisions.

6. PLOS authors have the option to publish the peer review history of their article (what does this mean?). If published, this will include your full peer review and any attached files.

Reviewer #1: No

Reviewer #2: No

---

## [Author Response · Author response to Decision Letter 0]

9 Sep 2024

Dear Cengiz Erisen, dear reviewers,

We would like to thank the two reviewers for carefully reading the manuscript and for their insightful and constructive feedback on the manuscript. We have tried to address all their critical observations. We believe that these revisions have notably improved the quality of our paper, and we hope you find the changes to be in line with the journal’s standards. We responded to each of the reviewers’ comments (in italic and indented). We have highlighted significant changes in yellow both in the manuscript and in the online appendix. Key revisions include additional model specifications with and without control variables, acknowledging and addressing certain theoretical insights highlighted by the reviewers, and editing the terminology. We sincerely thank you once again for considering this manuscript for publication.

With best regards!

Eylem Kanol & Rebecca Endtricht

Reviewer #1: “Conspiracy beliefs and negative attitudes towards outgroups in times of crises: Experimental evidence from Germany” is a manuscript (MS) that rests on a survey experiment administered to a large representative sample of the general German population consisting of 2,171 respondents. It investigates whether and to what extent exposure to conspiratorial framing of crisis scenarios impacts individuals' negative assessments of a diverse range of outgroups. The findings confirm that even vaguely suggestive vignettes implying conspiratorial thinking result in significant increase in hostility to certain outgroups like Muslims or Russians. I think the MS addresses an interesting topic with appropriate methods and mostly fine presentation. It can be published after a round of revision addressing the issues below.

The experiment results are first presented visually in figure 1 as relative group means. It is then reported that there are significant imbalances between groups in terms of pretreatment cofounders; age and migration background, so there is an OLS analysis where these are controlled for. But the same OLS controls for a number of other variables, too; for whose inclusion no similar justification is offered. Since this is a randomized experiment there is no legitimate point in controlling for additional variables, whose inclusion may generate suppressor effects on behalf of the treatments, potentially leading to bias.

So I think we need to see different OLS models where

i) the treatment categories are the only predictors

ii) age and migration would be included as a predictors alongside the treatment

iii) the current OLS model with the additional controls (to give a more comprehensive understanding of the correlates of outgroup hostility, rather than to isolate the effects of the treatment).

The 1st and 3rd of these may go to some sort of appendix to save space. Please also add a measure of fit to the models. (I believe I don’t have access to the current appendix that the MS mentions, so apologies if these are already present).

• We appreciate the reviewer's suggestion and have implemented it by estimating three models: one without controls, one with age and migration background as controls, and a final model that includes additional controls (also incorporating political party preference as suggested by Reviewer 2). In the main manuscript, we report the results from the second model, which includes the two control variables. The other two models are detailed in the Online Appendix. Additionally, we have included a measure of fit for each model. We now also use the Model 2 (our main model) for the analyses of the interaction effects. 

The moderating effects of ideology is an important topic that can be explored with more theoretical elaboration. Treatment effects look more certain and pronounced among the left compared to the right, which is a bit at odds with the theoretical expectations (“quadratic”) laid out in the lit rev section. So this can be discussed with a few sentences.

• In response to the reviewer's suggestion to further explore the role of ideology in our results, we have expanded our interpretation in the discussion section. While we acknowledge that the reviewer correctly observes that the treatment effects are somewhat more pronounced on the left, it is important to note that these effects are also significant and align with expectations on the right, but not among centrist respondents. This pattern supports the existence of a quadratic relationship, as discussed in the theory section. Therefore, we have retained our interpretation in this direction but also acknowledged how our findings contrast with some of the existing literature, which suggests a more linear relationship, particularly emphasizing the right's susceptibility to conspiracy theories.

The discussion of limitations is appropriate. However, the following sentence sounds like a non sequitur and I think should be dropped: “However, analyses of mean scores in the control group did not indicate a clear trend based on the order of the outgroups, suggesting that order effects were minimal. “Since we do not know the counterfactual of how baseline scores would fare under another random ordering we cannot infer from the observed scores whether the current ordering had any effect.

• We thank the reviewer for pointing this out. Indeed we cannot entirely rule out potential order effects. We have removed the sentence and adjusted the limitation accordingly by adding: “Therefore we cannot rule out the possibility that the order in which the outgroups were presented may have influenced respondents’ answers, potentially biasing the results.”

Minor: The MS has problems of precision in writing.

- At times the MS seems to display the strangely common academic mistake of conflating the concepts of “conspiracy theory” and “conspiracy”; and uses the latter to refer to the former. “Consequences of conspiracies” in the first sentence of abstract should be [consequences] “of conspiracy theories” or “of conspiracy beliefs” etc. instead. Similarly, the first subtitle that follows the intro should be sth like “conspiracy *theories* in times of crises” not conspiracies, since you are not examining conspiracies in any way.

• The reviewer is right in highlighting this mistake. We went through the manuscript to harmonize the terminology and to avoid the use of the term “conspiracy.” 

- “a quadratic relationship between the intensity of political ideology and conspiracy beliefs concerning a range of societal issues, noting that individuals at the ideological extremes are more inclined to endorse conspiracy theories”: Sounds like this should be “Quadratic relationship between the direction of (unidimensional) political ideology…” if extremes (i.e. those with more intense ideological beliefs, left or right) are indeed more inclined.

• We agree with the reviewer that the formulation was not clear. We have changed it accordingly so that it indicates that the respondents on the extremes are more inclined to endorse conspiracy theoris. 

- “heterogeneous effects of ideology” also sounds a misnomer. From the figures presented, ideology seems to have moderating effects for the treatment, resulting in heterogeneous effects of the *treatment*.

• Again we would like to thank the reviewer for their careful reading of the article and for pointing out this minomer. We have corrected this mistake and changed the title to: “Heterogeneous treatment effects across ideology.”

Reviewer #2: I find the manuscript titled "Conspiracy Beliefs and Negative Attitudes Towards Outgroups in Times of Crises: Experimental Evidence from Germany" to be a thought-provoking and engaging piece of research. The study looks into the consequences of conspiracy theory beliefs during crises, particularly in terms of intergroup attitudes. There is much to commend in this work, and it holds significant potential.

The article is very well-written, but some sections require further clarification. I recommend the following revisions:

1. Conceptual Clarification: The study focuses on the concept of "conspiracy beliefs" rather than "conspiracy mentality," which is a common focus in conspiracy theory research. However, the authors should clarify why they chose to use "conspiracy beliefs" instead of "conspiracy mentality." Is it because "conspiracy beliefs" are more suitable for experimental studies, or does the theoretical framework of the study necessitate this choice? A clear explanation of this decision would enhance the conceptual clarity of the paper.

• We thank the reviewer for their suggestion to clarify our terminology. We have included a brief definition of both “conspiracy belief” and “conspiracy mentality” in the section “Conspiracy theories in times of crises”, explaining why we use the term conspiracy belief. In our survey study, we do not directly measure “conspiracy mentality,” which is conceptualized as a personality trait, but instead focus on conspiracy beliefs in specific (experimentally induced) situations.

2. Literature on the Consequences of Conspiracy Beliefs: While the sections “Conspiracy Theories in Times of Crises” and “Negative Outgroup Attitudes as a Consequence of Conspiracies” briefly address the literature, the paper would benefit from a more comprehensive discussion of the consequences of conspiracy beliefs. The field is rich with studies on this topic, and incorporating more of this literature could strengthen the paper. A dedicated section on the consequences of conspiracy beliefs, placed between the two aforementioned sections, could help readers better understand the study's significance.

• This is also a very good suggestion by the reviewer. We have included a discussion of recent research on consequences of conspiracy beliefs on the societal level, including consequences regarding social, health-related, and political circumstances and outcomes. However, given the extensive nature of the existing research, we are unable to address all areas in sufficient detail within our manuscript. Thus, we have further referenced two comprehensive reviews on this topic in our manuscript. We hope that this brief overview aligns with the reviewer's expectations. 

3. Ideological Extremity and Conspiracy Beliefs: The study’s findings supporting the ideological extremity hypothesis are insightful in understanding the role of ideologies in conspiracy beliefs. However, the paper should also discuss alternative perspectives on the role of ideology in conspiracy beliefs. Citing literature that suggests right-wing supporters are more prone to conspiracy theories, or that there are no significant ideological differences in endorsing conspiracy theories, could provide a more balanced view and inform readers about different arguments on this topic.

• We appreciate the reviewer’s constructive feedback. In line with their suggestion, we enhanced the literature review and emphasized existing studies that show how right-wing supporters are more prone to conspiracy theories. We also point to a recently published study which employed a more nuanced research design and highlighted how the endorsement of conspiracy theories depends on their specific content and their political endorsment by partisan/ideological elites (p. 10). 

4. Research Design and Data Section: The authors note that 12% of respondents self-identified as right-wing, compared to 43% identifying as left-wing and 44% as centrist. This distribution is intriguing and warrants further discussion. What might explain these differences? Could it be the survey sampling strategy, or perhaps it reflects the current political climate in Germany? Addressing these questions could provide valuable context for interpreting the findings.

• We added a brief statement discussing this distribution along with additional references. At the time of the survey left and center-left parties were very strong in Germany, which may partially explain why many respondents self-placed themselves on the left side of the scale. Moreover, previous studies indicate that such a skewed distribution is quite typical for Germany, when it comes to ideological self-placement. Nevertheless we now highlight this limitation and stress the potential under-representation of right-wing supporters in the sample.

5. Control Variables: The study controls for gender, level of education, religion, migration background, form of survey administration, and whether respondents resided in a former East German state. However, the authors might consider including party identity as a control variable, especially given the rise of the Alternative for Germany (AfD) and its influence on German politics. Including party identity (or vote choice) could offer additional insights, particularly considering the party's stance on migration.

• This is a good suggestion by the reviewer. In the survey the respondents were asked, which party they would vote for if the German federal election was taking place next Sunday. We used this variable as a proxy for party identity and included it in the Model 3 (see also the discussion above). Including this variable did not change our findings. 

Overall, I found this manuscript to be an engaging read, and I believe it could make a significant contribution to the field after some revisions.

---

## [Decision Letter · Decision Letter 1]

7 Oct 2024

Conspiracy beliefs and negative attitudes towards outgroups in times of crises: Experimental evidence from Germany

PONE-D-24-24527R1

Dear Dr. Kanol,

We’re pleased to inform you that your manuscript has been judged scientifically suitable for publication and will be formally accepted for publication once it meets all outstanding technical requirements.

Kind regards,

Cengiz Erisen

Academic Editor

PLOS ONE

Additional Editor Comments (optional):

Reviewers' comments:

Reviewer's Responses to Questions

**Comments to the Author**

1. If the authors have adequately addressed your comments raised in a previous round of review and you feel that this manuscript is now acceptable for publication, you may indicate that here to bypass the “Comments to the Author” section, enter your conflict of interest statement in the “Confidential to Editor” section, and submit your "Accept" recommendation.

Reviewer #1: All comments have been addressed

Reviewer #2: All comments have been addressed

2. Is the manuscript technically sound, and do the data support the conclusions?

Reviewer #1: Yes

Reviewer #2: Yes

3. Has the statistical analysis been performed appropriately and rigorously? 

Reviewer #1: Yes

Reviewer #2: Yes

4. Have the authors made all data underlying the findings in their manuscript fully available?

Reviewer #1: Yes

Reviewer #2: Yes

5. Is the manuscript presented in an intelligible fashion and written in standard English?

Reviewer #1: Yes

Reviewer #2: Yes

6. Review Comments to the Author

Reviewer #1: I find the revisions satisfactory and congratulate the authors for an interesting article. I think the manuscript can now be published.

Reviewer #2: Thank you for your thoughtful responses to my initial comments and for the revisions made to the manuscript. I appreciate the careful consideration you have given to each point, and the updated manuscript reflects a stronger theoretical and empirical foundation.

Overall, I find the paper much improved and believe it makes a significant contribution to the study of conspiracy beliefs, their consequences, and their ideological contexts. I would like to note a minor additional comments for your consideration:

Wording Clarification on p.11: In the sentence, "Similarly, extremist political extremists frequently target identifiable 'evil' groups in their rhetoric," there seems to be a potential wording error with the repetition of "extremist." It appears that the intended phrase is "political extremists," rather than "extremist political extremists." Clarifying this would enhance the readability and accuracy of the statement.

Beyond this minor point, I have no further suggestions. The clarifications provided on the conceptual focus, the expanded discussion on the consequences of conspiracy beliefs, the balanced perspectives on the role of ideology, and the additional context on the distribution of political self-placement all enhance the manuscript's clarity and depth. Furthermore, the decision to include party identity as a control variable is an excellent addition and helps address potential nuances in the data.

7. PLOS authors have the option to publish the peer review history of their article (what does this mean?). If published, this will include your full peer review and any attached files.

Reviewer #1: No

Reviewer #2: No

---

## [Editor Report · Acceptance letter]

18 Oct 2024

PONE-D-24-24527R1 

PLOS ONE

Dear Dr. Kanol, 

I'm pleased to inform you that your manuscript has been deemed suitable for publication in PLOS ONE. Congratulations! Your manuscript is now being handed over to our production team.

Kind regards, 

on behalf of

Dr. Cengiz Erisen 

Academic Editor

PLOS ONE